# The Effects of Annealing at Different Temperatures on Microstructure and Mechanical Properties of Cold-Rolled Al$_{0.3}$CoCrFeNi High-Entropy Alloy

**Zichao Zhu** [1,2]**, Tongtong Yang** [1,2]**, Ruolan Shi** [3]**, Xuantong Quan** [1,2]**, Jinlong Zhang** [1,2]**, Risheng Qiu** [1,2,*]**, Bo Song** [3,*] **and Qing Liu** [1,4]

1    International Joint Laboratory for Light Alloys (MOE), College of Materials Science and Engineering, Chongqing University, Chongqing 400044, China; 20182950@cqu.edu.cn (Z.Z.); 20183072@cqu.edu.cn (T.Y.); qxt18875240382@163.com (X.Q.); 20152739@cqu.edu.cn (J.Z.); qingliu@cqu.edu.cn (Q.L.)
2    Electron Microscopy Center, College of Materials Science and Engineering, Chongqing University, Chongqing 400044, China
3    School of Materials and Energy, Southwest University, Chongqing 400715, China; srl666@email.swu.edu.cn
4    Key Laboratory for Light-Weight Materials, Nanjing Tech University, Nanjing 210009, China
*    Correspondence: rsqiu@cqu.edu.cn (R.Q.); bosong@swu.edu.cn (B.S.)

**Abstract:** In this work, cold-rolling was utilized to induce a high density of crystal defects in Al$_{0.3}$CoCrFeNi high-entropy alloys. The effects of annealing temperature on static recrystallization, precipitation behavior and mechanical properties were investigated. With increasing annealing temperature from 590 °C to 800 °C, the area fraction of recrystallized region increases from 26.9% to 93.9%. Cold-rolling deformation largely promotes the precipitation of B2 phases during annealing, and the characteristics of the precipitates are linked to recrystallization level. The coarse and equiaxed B2 phases exist in the recrystallized region and the fine and elongated B2 phases occupy the non-recrystallized region. Combined use of cold-rolling and annealing can remarkably enhance the strength and toughness. A partially recrystallized microstructure in a cold-rolled sample annealed at 700 °C exhibits a better combination of strength and toughness than a fully recrystallized microstructure in a cold-rolled sample annealed at 800 °C. Finally, related mechanisms are discussed.

**Keywords:** Al$_{0.3}$CoCrFeNi; cold-rolling; precipitation; recrystallization; mechanical properties

## 1. Introduction

High-entropy alloys (HEAs) contain five or more elements at concentrations of 5–35%, and thus exhibit significant multi-element effects, e.g., high entropy, lattice distortion, sluggish diffusion and cocktail effects [1,2]. The effects generate anomalous properties in HEAs, such as high strength, good ductility, excellent fracture toughness and outstanding resistances to corrosion, radiation and fatigue [3,4]. Thus, HEAs with multiple-principal elements have received extensive attention and are also considered as potential candidates for engineering materials in the automotive, aerospace and marine fields.

The HEAs tend to form simple solid solution phases, such as face-centered cubic (FCC), body-centered cubic (BCC) or hexagonal close-packed (HCP) structures [5–8]. CoCrFeMnNi alloy is a typical FCC HEA and its main disadvantage is the low yield stress [9]. Work hardening and refining hardening can be employed to increase the yield strength of HEA alloys [10,11]. Moreover, alloying is also an effective method to increase the yield stress by solid solution hardening or precipitation hardening [12–16]. The FCC Al$_{0.3}$CoCrFeNi is the most widely studied precipitation hardening HEA alloy [16–20]. It has been reported that two stable phases (L1$_2$ and B2 phases) with order structure can be favorably precipitated within the FCC matrix [18,21,22]. With increasing temperature, phase transformations from FCC phase to a L1$_2$ phase and finally to a B2 phase can be observed in as-cast alloys [18]. The L1$_2$ phase can be homogeneously precipitated within FCC matrix, but it has a low

barrier effect on slip. In contrast, the B2 phase can generate stronger hardening effect than the $L1_2$ phase. However, the B2 phase has a high nucleation barrier and is only precipitated at above 700 °C in as-cast alloys [21]. Pre-inducing crystal defects are expected to increase the nucleation sites for precipitation. Tang et al. [23] found that refining grains can promote the precipitation of the B2 phase. In the nanocrystalline HEA, direct transformation from the FCC phase to the B2 phase was observed during annealing at 320 °C. Wang et al. [24] used pre-tensile deformation (a strain of 10%) to promote the nucleation of the B2 phase during annealing at 700 °C. Moreover, cold-rolling was also usually employed to tailor the precipitation behavior and microstructure [18–20,22]. It has been found that profuse dislocations and some deformation twins can be formed in the $Al_{0.3}CoCrFeNi$ matrix during cold-rolling. These defects via cold-rolling can also promote the precipitation of the B2 phase during subsequent annealing, leading to an increase in strength. In addition to the precipitation, recovery and recrystallization during annealing can also affect the mechanical properties [20]. It has been reported that the amount of recrystallization is typically dependent on annealing temperature and independent of time [20–25]. Thus, annealing treatment at various temperatures might influence the recrystallization level of cold-rolled $Al_{0.3}CoCrFeNi$ alloys. However, few studies have focused on the effect of recrystallization level on the precipitation behavior and mechanical properties in cold-rolled $Al_{0.3}CoCrFeNi$ alloys. In the present work, cold-rolling is used to induce a severely deformed microstructure. Annealing treatment at different temperatures was carried out to investigate the precipitate behavior and recrystallization level. Finally, the mechanical properties were evaluated.

## 2. Materials and Methods

The HEA ingot with a nominal composition of $Al_{0.3}CoCrFeNi$ was prepared by the vacuum arc re-melting method. Commercial grade pure elements of Al-99.9%, Co-99.9%, Cr-99.9%, Fe-99.9% and Ni-99.9% were used as raw materials. The ingots were flipped and re-melted at least four times to ensure the homogeneity of chemical composition. The dimensions of the solidified ingot were about 80 mm × 80 mm × 130 mm. The as-cast sample with homogenized microstructure was marked as AH sample. The plates with a dimension of 30 mm (RD) × 40 mm (TD) × 15 mm (ND) cut from the AH sample was cold-rolled by a thickness reduction of 90%. The RD, TD and ND represent the rolling direction, transverse direction and normal direction of the plates, respectively. Annealing treatment at various temperatures was carried out to induce recrystallization and precipitation. Dasari et al. [20] found that after the annealing time exceeded 4 h, the amount of recrystallization showed little change in cold-rolled $Al_{0.3}CoCrFeNi$ high-entropy alloys. Thus, the annealing time was set to 10 h or more. In order to induce sufficient precipitation, for annealing at below 700 °C, the annealing time was set to 24 h, and for annealing at above 700 °C, the annealing time was set to 10 h [20–26]. Detail processing history is listed in Table 1.

**Table 1.** Rolled Al0.3CoCrFeNi sheets subjected to various processing histories.

| Sample | Processing History |
| --- | --- |
| AH | The as-cast sample |
| AR | AH sample cold-rolled by a thickness reduction of 90% |
| AR590 | AR sample annealed at 590 °C for 24 h |
| AR660 | AR sample annealed at 660 °C for 24 h |
| AR700 | AR sample annealed at 700 °C for 10 h |
| AR800 | AR sample annealed at 800 °C for 10 h |

X-ray diffractometer (XRD, Rigaku D/MAX2500PC, RigakuCo., Tokyo, Japan) was used to analyze the phase structure, and the copper target was selected as the diffraction target. The working voltage of the equipment was 40 KV, the scanning speed was 4 °/min and the step width was 0.02°. XRD patterns were analyzed by the PDF2-2004 database. The microstructures of the $Al_{0.3}CoCrFeNi$ specimens were examined by scanning electron

microscope (SEM, TESCAN's MIRA 3, Brno, Czech Republic) and electron backscattering diffraction (EBSD, Oxford AZtech Max2, Oxford Instruments, London, UK). TECNAIG2 F20 transmission electron microscope (TEM, FEI, Hillsboro, USA) with a working voltage of 200 KV was used to analyze the morphology, structure and composition of HEA alloys. Dog-bone-shaped specimens with gauge dimensions of 10 mm (length) × 4 mm (width) × 11 mm (thickness) were prepared for tensile test along the RD. The tensile tests were performed on an LD26.105 material test machine (LiShi (Shanghai) Instruments Co., Ltd., Shanghai, China) at a constant strain rate of $1 \times 10^{-3}$ s$^{-1}$. Each mechanical test was repeated at least three times to get representative results.

## 3. Results

Figure 1a shows the SEM image of the AH sample. It was found that the AH sample has a single-phase structure with coarse grain size (>100 μm). The TEM bright-field image of the AH sample (see Figure 1b) indicates that the overall contrast is relatively uniform, and no second phase is found. Moreover, some scattered dislocation lines can also be found within grains. After cold-rolling ~90%, initial coarse grains were deformed and broken, resulting in that the shape of grains are squashed into a fibrous deformation microstructure, as shown in Figure 1c. The TEM image in Figure 1d indicates that a high density of dislocations is formed during cold-rolling. Moreover, some deformation twins can also be found in the AR sample (see the red arrow in Figure 1d). Such deformation twins have been observed in the deformed Al$_{0.3}$CoCrFeNi alloys by cold-rolling or tension at room temperature. This is attributed to the relatively low stacking fault energy [27]. Figure 1e shows the XRD patterns of two alloys, confirming that the AH alloy has a single FCC phase without precipitates, and cold-rolling at room temperature retains such a single FCC phase structure.

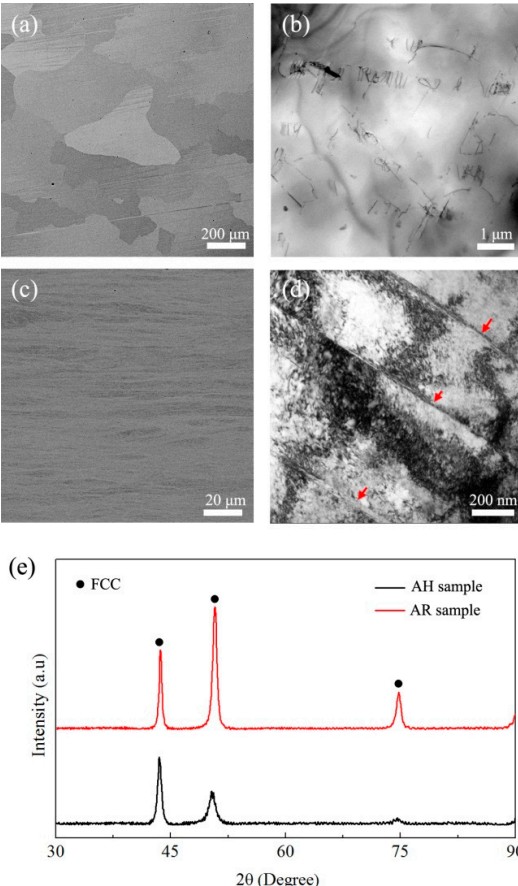

**Figure 1.** (**a**) SEM image of AH sample; (**b**) TEM image of AH sample; (**c**) SEM image of AR sample; (**d**) TEM image of AR sample; (**e**) XRD patterns of various samples.

Figure 2 shows the EBSD maps of cold-rolled and annealed samples, indicating that annealing causes the evolution of low angle boundaries (LABs) in the AR sample. The density of LABs (the length of the LABs per area) is calculated and listed in Table 2. After cold-rolling, a high density of LABs (~0.265 $\mu m^{-1}$) can be found within grains, which is related to profuse dislocations [28]. Annealing treatment can remarkably reduce the density of LABs owing to static recovery and static recrystallization. When the annealing temperature is lower than 700 °C, a hetero-structure can be formed, which contains the recrystallized region (RS) and the non-recrystallized region (NRS). Figure 2b shows the EBSD map of the AR660 sample. In the recrystallized regions, FCC matrix exhibits equiaxed fine grains (~1.25 $\mu m$) and very low LAB density (~0.011 $\mu m^{-1}$). Some non-recrystallized grains exhibit a fiber shape elongated along the RD. This indicates that a partial recrystallized microstructure can be generated when the annealing temperature is lower than 700 °C. The LAB density of the non-recrystallized region in the AR660 sample (~0.141 $\mu m^{-1}$) is lower than in the AR sample, owing to static recovery. With increasing annealing temperature to 800 °C, the density of LABs reduces to 0.009 $\mu m^{-1}$, and full recrystallization remarkably refines grains to form a uniform microstructure with an average grain size of ~1.31 $\mu m$. It was also found that annealing temperature exhibits little influence on recrystallized grain size. This might be attributed to the strong pining effect of the B2 phases on the movement of the boundary motion required for recrystallization growth [20–29]. Moreover, annealing twins can be formed within the recrystallized grains, which are identified as ~60° Σ3 twins. The density of Σ3 boundary is also calculated and listed in Table 2. It was also found that annealing temperature has little influence on density of Σ3 twin boundaries within recrystallized grains. Moreover, a small amount of Σ3 twin boundaries can also be formed in the non-recrystallized grains.

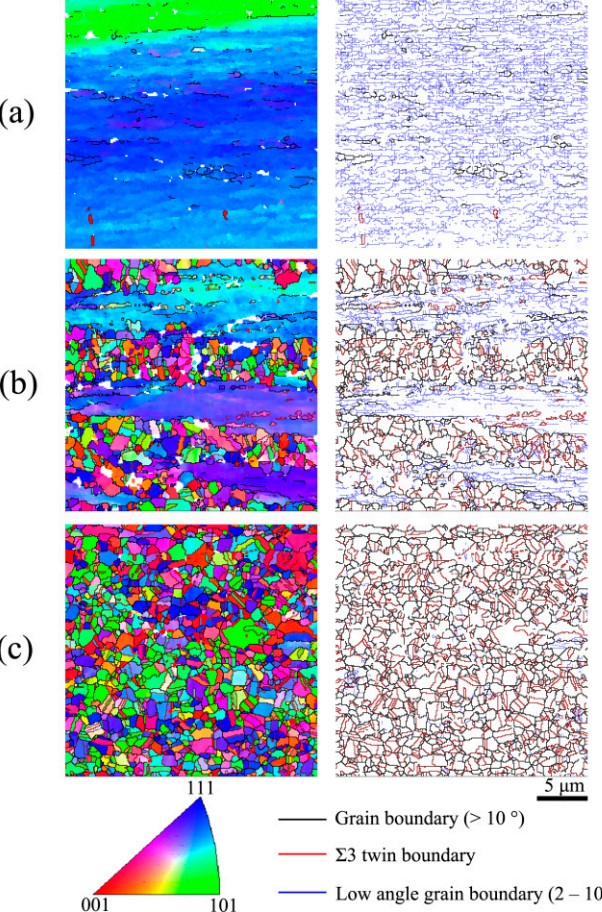

**Figure 2.** EBSD maps of (**a**) AR sample, (**b**) AR660 sample and (**c**) AR800 sample.

**Table 2.** Length of LABs (2–10°) per area and Length of Σ3Bs per area.

| Sample | LABs ($\mu m^{-1}$) | Σ3Bs ($\mu m^{-1}$) |
|---|---|---|
| AR | 0.265 | 0.001 |
| AR660-RS | 0.011 | 0.031 |
| AR660-NRS | 0.141 | 0.007 |
| AR800 | 0.009 | 0.034 |

Figure 3 shows the SEM images of AR alloys annealed at various temperatures, indicating that annealing induces a large number of particles, and the feature of precipitate is largely dependent on the degree of recrystallization. Similar to Figure 2, when the annealing temperature is lower than 700 °C, the microstructure usually contains a recrystallized region and a non-recrystallized region. In the recrystallized region, the precipitates have a coarse size. In contrast, precipitates within the non-recrystallized region are finer. Figure 4 shows SEM images of the AR600 sample with high magnification. In the AR600 sample, the fine-scale elongated precipitates in the non-recrystallized region decorate the deformation bands. In the recrystallized region, coarse precipitates are usually located at the grain boundaries and some fine precipitates can also be found within recrystallized grains. The mean area per particle ($A_p$) and area fraction of particles ($A_f$) are summarized by five SEM images with an area of 18,496 $\mu m^2$, which are listed in Table 3. This clearly shows that recrystallized regions have larger $A_P$ and $A_f$ than non-recrystallized regions. With increasing annealing temperature to 800 °C, the equiaxed precipitates are evenly distributed in the whole matrix due to the occurrence of complete recrystallization. Table 3 also indicates that the annealing temperature has little influence on $A_P$ and $A_f$ in recrystallized regions.

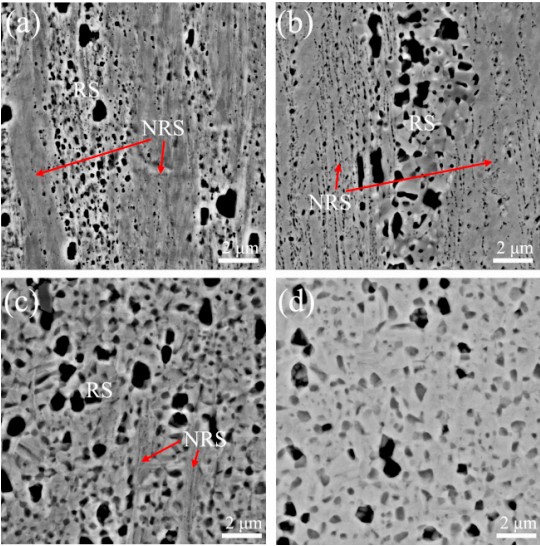

**Figure 3.** SEM-ECC images of AR sample aged at various temperatures. (**a**) AR590 sample; (**b**) AR660 sample; (**c**) AR700 sample; (**d**) AR800 sample.

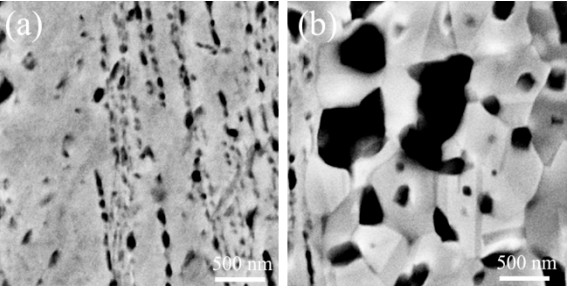

**Figure 4.** SEM images of AR660 sample (**a**) non-recrystallized regions; (**b**) recrystallized regions.

**Table 3.** The mean area per particle and area fraction of particles in various samples.

| Sample | AR660 | | AR800 |
|---|---|---|---|
| | RS | NRS | |
| $A_p$ (μm$^2$) | $0.018 \pm 0.002$ | $0.005 \pm 0.003$ | $0.019 \pm 0.001$ |
| $A_f$ (%) | $35.2 \pm 5.1$ | $19.9 \pm 3.3$ | $32.1 \pm 2.1$ |

Clearly, the recrystallization level will influence the distribution of precipitates with different size. The area fraction of the recrystallized region was calculated by using five SEM images with an area of 18,496 μm$^2$ and listed in Table 4. For the AR590 sample, the area fraction of the recrystallized region is only 26.9%. With increasing annealing temperature, the area fraction of the recrystallized region increases. After annealing at 800 °C for 10 h, almost a fully recrystallized microstructure can be obtained.

**Table 4.** Mean area fraction of recrystallized regions in various samples.

| Sample | AR | AR590 | AR660 | AR700 | AR800 |
|---|---|---|---|---|---|
| $f_{RS}$ | 0% | $26.9\% \pm 0.3\%$ | $74.3\% \pm 0.7\%$ | $87.7\% \pm 0.2\%$ | $95.9\% \pm 0.3\%$ |

To further reveal the feature of precipitate, the TEM images of the AR590 sample are shown in Figure 5. Numerous micro-sized precipitates are observed in the FCC grains in Figure 5a,b. The Al-elemental energy dispersive spectrum of the region indicated by red dashed line in Figure 5b is shown in Figure 5c. The (Al-Ni)-rich B2 precipitate can be clearly identified in Figure 5c. Apparently, the B2 precipitates possess a dominating proportion in the AR590 sample. Figure 5d exhibits the morphology, composition and BCC structure of the B2 precipitate. The composition of the B2 precipitation acquired by TEM equipped with energy dispersive spectroscopy is Al=19.44%, Cr = 2.05%, Fe = 11.3%, Co = 12.44%, Ni = 54.77% in weight percentage. The SADP recorded from the B2 precipitate (see Figure 5d) can be indexed firmly as the {111} bcc zone axis (ZA). This indicates that the B2 phase is an ordered BCC structure. In addition, the Kurdjumov–Sachs (K-S) relationship between B2 and the FCC matrix was reported, which is that the {110}$_{FCC}$ is parallel to {111}$_{B2}$ while the {111}$_{FCC}$ is parallel to {110}$_{B2}$ [20]. Besides the Al-Ni rich second phase, the other two precipitates (Cr-rich BCC precipitate and Cr-Fe-Co-rich σ precipitate) were detected in AR590 specimen [22]. The BCC phase consists of Al = 1.04%, Cr = 84.44%, Fe = 7.23%, Co = 4.39% and Ni = 2.9% in weight percentage. SADP shown in left bottom of Figure 5e proves that the BCC phase has a BCC structure which is consistently indicted as the {110}bcc ZA. In contrast, the (Cr-Fe-Co)-rich σ phase is composed of elements Al = 0.17%, Cr = 51.51%, Fe = 20.65%, Co = 22.47% and Ni = 5.19% in weight percentage. The σ phase is a tetragonal structure, which is speculated by the bright field transmission electron microscope image with the corresponding {001} ZA diffraction pattern, as shown in Figure 5f. Figure 6 shows the STEM images and elemental energy spectra of the precipitates in the AR800 sample. Accompanied by the increase in annealing temperature, the size of the B2 precipitates in the AR800 sample is slightly larger than in the AR590 sample, as shown in Figures 5c and 6b. With increasing annealing temperature, BCC phases disappear and the number of σ phases decreases. Furthermore, part of the (Al-Ni)-rich B2 phase and the (Cr-Fe-Co)-rich σ phase are attached to each other to nucleate, as indicated by the red arrow in Figure 6c,d. It is able to account for the formation of the two precipitates from the element diffusion perspective. Previous studies have shown that the enriched elements of the B2 phase and the σ phase particles are opposite, making the two phases mutually promote precipitation and growth [20].

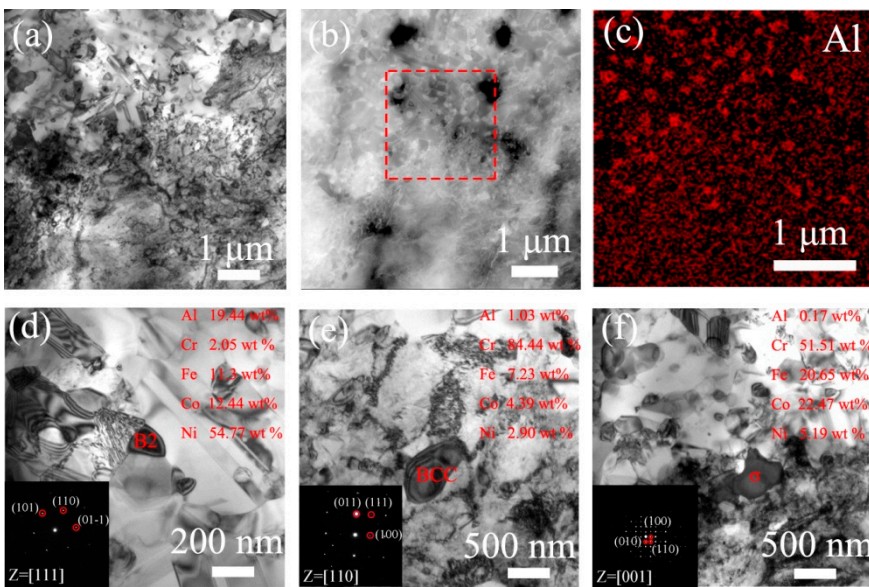

**Figure 5.** TEM images of AR590 sample: (**a**) BF image, (**b**) STEM image, (**c**) elemental energy dispersive mapping, (**d**) B2 precipitate with SADP and composition, (**e**) BCC precipitate with SADP and composition, (**f**) σ precipitate with SADP and composition.

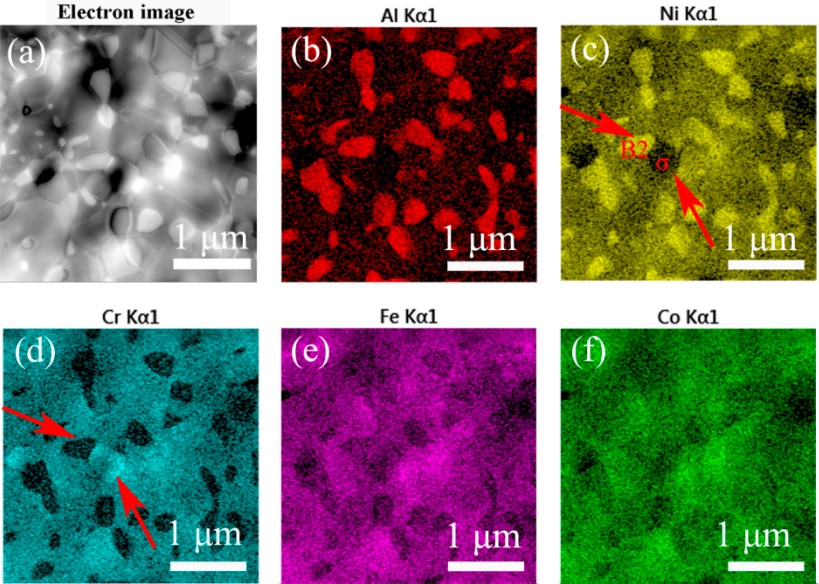

**Figure 6.** (**a**) STEM images in AR800 sample and corresponding elemental mapping of (**b**) Al, (**c**) Ni, (**d**) Cr, (**e**) Fe, (**f**) Co.

Based on the above analysis, it can be seen that in the annealing temperature range of 590 °C to 800 °C, the B2 phase is the main precipitate. Figure 7 compares the chemical composition of the B2 phase in the samples annealed at various temperatures. It was found that annealing temperature has little influence on the chemical composition of the B2 phases. TEM bright field images of the AR800 sample are used to further reveal the nucleation and distribution of B2 precipitates, as shown in Figure 8. The coarse B2 phases are favorably precipitated at the grain boundaries. Moreover, fine B2 phases are also precipitated within recrystallized grains and at the annealing twin boundaries. This is consistent with the results in Figure 4. Moreover, Figure 6b shows that the content of Al and Ni elements in the B2 phases within grains are lower than in the B2 phases at grain boundaries.

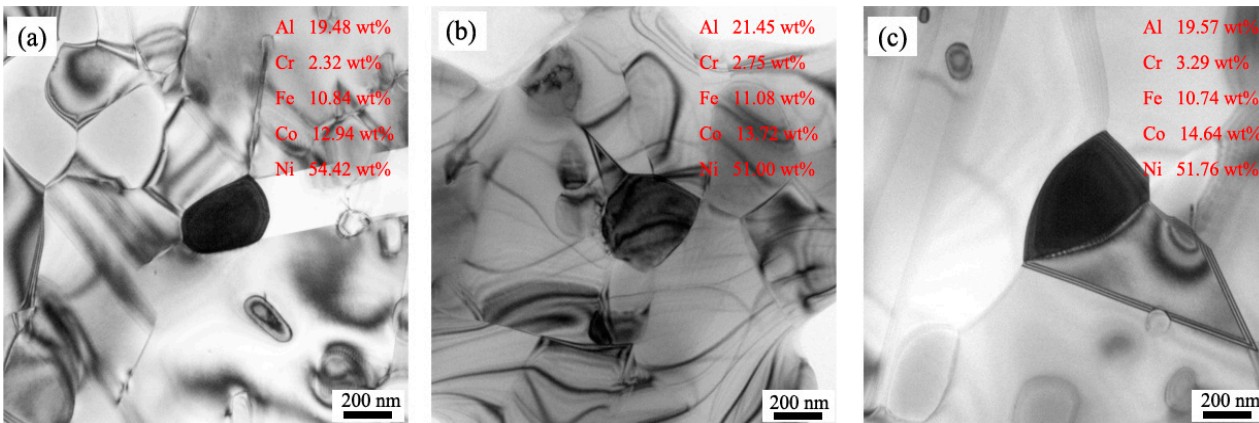

**Figure 7.** B2 phase images of various samples: (**a**) AR600 sample, (**b**) AR700 sample, (**c**) AR800 sample.

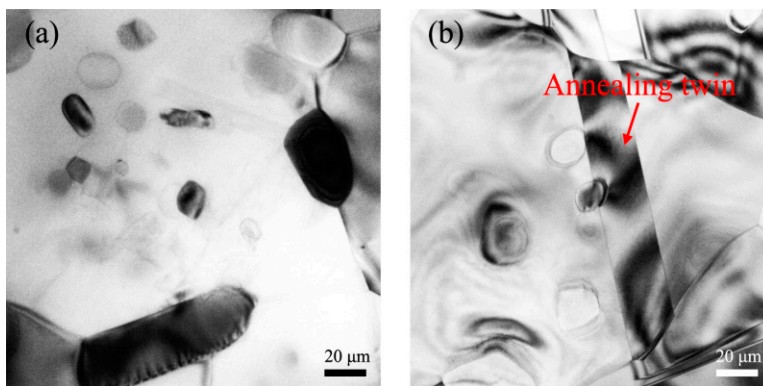

**Figure 8.** TEM images of AR800 sample. (**a**) B2 phases at grain boundaries and within grains; (**b**) B2 phases at twin boundaries.

The tensile curves of the various samples are shown in Figure 9. The mechanical properties are listed in Table 5. The AH sample with a single FCC phase exhibits very low yield strength (~170 MPa) and large uniform elongation (~43.3%). Moreover, the AH sample exhibits a very low strain hardening rate in the beginning of deformation. This behavior is similar to that of low stacking fault energy alloys (e.g., CoCrFeMnNi HEA, TWIP steel et.) [30,31]. Cold-rolling remarkably enhances the yield strength to 1060 MPa. However, the uniform elongation is reduced to 0.4%. The increase in yield strength in the AR sample can be attributed to the high work hardening effect via cold-rolling. The strong barrier effect on the movement of dislocations leads to a very high strain hardening rate, which rapidly increases stress to peak strength of 1273 MPa. Annealing treatment induces the precipitation of second phases. It is considered that the B2 phase within grains can generate an Orowan hardening effect on the yield strength. However, annealing treatment reduces the yield strength of the AR sample. This can be mainly attributed to the softening effect via static recovery and recrystallization. The AR590 sample shows the deformation characteristics of brittle materials with almost no tensile ductility (~0.1%). With increasing annealing temperature to 660 °C, the tensile ductility is increased to 19.4%. Further increasing the annealing temperature can reduce the yield strength, but has a smaller effect on tensile ductility. For the AR800 sample, the work hardening effect via cold-rolling is fully eliminated and the yield strength is reduced to 837 MPa.

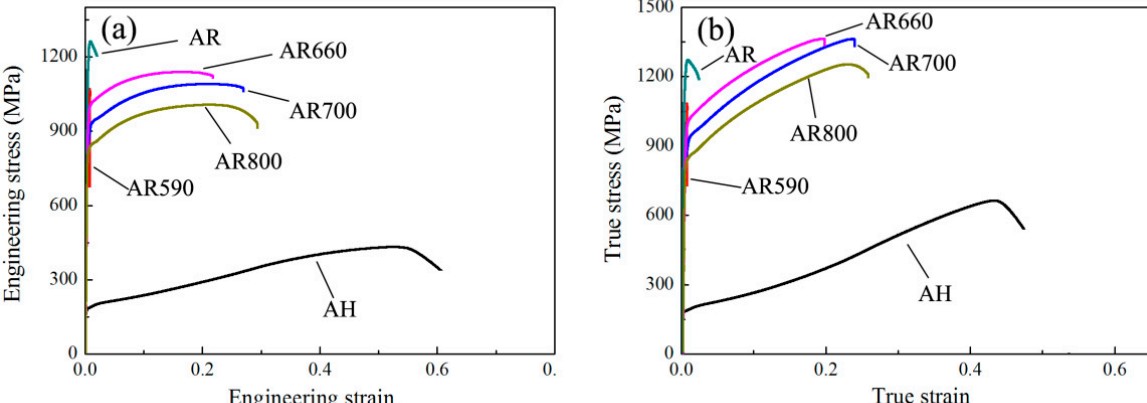

**Figure 9.** (**a**) Engineering stress vs. engineering strain curves and (**b**) true stress vs. true strain curves of various samples.

**Table 5.** Yield strength (YS), peak strength (PS) and uniform elongation (UE) of various samples.

| Sample | YS (MPa) | PS (MPa) | UE |
|--------|----------|----------|-----|
| AH | 170 | 659 | 43.3% |
| AR | 1060 | 1273 | 0.4% |
| AR590 | 995 | 1065 | 0.1% |
| AR660 | 1020 | 1366 | 19.4% |
| AR700 | 939 | 1362 | 23.3% |
| AR800 | 837 | 1248 | 22.7% |

## 4. Discussion

### 4.1. Influence of Cold-Rolling on Precipitation Behavior

The mole fraction of equilibrium phases at various temperatures has been reported in previous work [18,22]. In the temperature range of 590 °C to 800 °C, the equilibrium phase composition is FCC+B2+σ. However, the amount of (Cr-Fe-Co)-rich σ precipitate is far lower than that of the (Al-Ni)-rich B2 precipitate, as shown in Figures 5 and 6. This can be attributed to the fact that the mixing enthalpy of Ni and Al is much negatively lower than the other atom pairs of the five principal elements [16]. Thus, the B2 phase is the main precipitate for this temperature range. In fact, homogenous distribution of $L1_2$ phases usually dominate the FCC matrix in annealed as-cast alloys [21]. This unexpected precipitation of a metastable $L1_2$ phase can be widely found in the as-cast alloy annealed at 750 °C [32]. This has been attributed to the fact that the nucleation barrier of the $L1_2$ phase is much lower than that of the B2 phase [33,34]. The underlying reason is that the FCC/B2 interface energy is higher than the FCC/$L1_2$ phase [35]. In the present work, it is found that cold-rolling can largely promote the nucleation of the B2 phase. It is considered that the high-density crystal defects (i.e., dislocations, twins, etc.) produced by cold-rolling provide more heterogeneous nucleation sites for the B2 phase [20,22–24]. Moreover, grain boundaries are important heterogeneous nucleation sites for B2 precipitates. Thus, the fine grain size in the annealed AR samples is also benefit for the precipitation of the B2 phase.

Figure 3 shows that recrystallization level influences the distribution of B2 precipitates with different sizes. It was found that recrystallized regions and non-recrystallized regions exhibit distinct precipitation characteristics. In the non-recrystallized region, B2 phases exhibit fine size and elongated morphology. A previous report has revealed that the deformation and twin bands in rolling deformed regions can provide high diffusivity of Al and Ni along the twin boundaries to precipitate the B2 phase [20]. It is considered that the thickness of the B2 precipitate in deformation and twin bands might be limited. Thus, B2 phases in the non-recrystallized region exhibit an elongated morphology. In the recrystallized region, B2 phases usually nucleate at the grain boundaries and have a coarse size and equiaxed morphology. This indicates that the grain boundaries provide a pathway for rapid long-distance elemental diffusion to promote the nucleation and growth of B2

phases. The formation of annealing twin boundaries in the recrystallized region provides more nucleation sites for B2 phases. Moreover, some fine and equiaxed B2 phases can also be found within recrystallized grains, as shown in Figure 8. They may be nucleated in the regions with high local density of dislocations [20]. With the growth of recrystallized grains, these B2 phases evolve into within grains. The B2 phases within grains exhibit smaller size and lower content of (Al, Ni) than those at the grain boundaries. This may be attributed to the fact that the element atoms are difficult to migrate within grains due to the delayed diffusion effect.

### 4.2. Mechanical Properties

The combined use of cold-rolling and annealing can remarkably enhance the yield strength. After cold-rolling, a high density of dislocations and nano-twins can generate dislocations hardening and refinement hardening via twin boundaries, respectively [36]. Thus, significant high yield strength can be obtained in the AR sample. However, the high density of crystal defects largely deteriorates tensile ductility. Annealing treatment can precipitate secondary phase and arouses recovery and recrystallization. The former can generate precipitation hardening to enhance the yield strength and reduce tensile ductility. The latter can weaken work hardening effect via cold-rolling, resulting in a decrease in yield strength [22]. It was found that annealing at 590 °C simultaneously reduces the yield strength and tensile ductility. The poor ductility might be attributed to the high area fraction of the non-recrystallized region (~73.1%) and precipitation of second phases, as shown in Figure 3 and Table 4.

With increasing area fraction of recrystallized regions, tensile ductility can be improved. After annealing at 800 °C, a fully recrystallized microstructure can be obtained. Compared to the AR sample, the uniform ductility is largely improved from 0.4% to 22.7%, as shown in Table 5. In the AR800 sample, the work hardening via cold-rolling is almost completely eliminated. However, the AR800 sample (~837 MPa) exhibits far higher yield strength than the AH sample (~170 MPa). As shown in Figure 2, a remarkable grain refinement can be achieved in the AR800 sample. Kumar et al. [37] developed a Hall–Petch type equation ($\sigma_{HP} = 174 + \frac{371}{\sqrt{d}}$) to reveal the relationship between the yield strength and grain size in a single FCC $Al_{0.1}CoCrFeNi_{0.5}$ alloy. The AR800 sample has an average grain size ~1.31 μm, the Hall–Petch strengthening can be estimated to be ~498 MPa. Thus, it can be assumed that 837 MPa (yield strength) −498 MPa (Hall–Petch hardening) = 339 MPa is the precipitation strengthening contribution due to the B2 precipitates. Moreover, it is considered that Σ3 twin boundaries can also subdivide grains to generate refinement hardening contribution [38].

It is interesting to find that the AR700 sample with a partial recrystallized microstructure has close tensile ductility and higher yield strength than the AR800 sample. Static toughness of various samples is also calculated by true stress-strain curves and listed in Table 6. This indicates that partial recrystallized microstructure in the AR700 sample exhibits a better combination of strength and toughness. In fact, annealing temperature exhibits little influence on recrystallized grain size. The high yield strength can be attributed to the high work hardening effect in the non-recrystallized regions. For AR700, a high tensile ductility and toughness might also be attributed the effect of heterogeneous structure. Several studies on heterogeneous microstructure have been widely reported [18,39–41]. A reasonable configuration of heterogeneous structure can usually lead to a remarkable combination of strength and ductility [38–40]. It has been reported that strain incompatibility in heterogeneous microstructure can generate back stress strengthening [42]. High magnitude of back stress has been reported in $AlCoCrFeNi_{2.1}$ alloys [43]. Moreover, the back stresses generated in heterogeneous microstructures can also enhance the strain hardening ability [42]. Kim et al. [44] found high strength and high ductility in as-rolled CoCrFeCuNi alloys are related to the back-stress generated by heterogeneous lamella structure. Sathiyamoorthi et al. [18] reported that the back stress strengthening via strain heterogeneity resulted in good strength-ductility synergy effect in an Al0.3CoCrNi

medium-entropy alloy. This may be the reason why the AR700 samples exhibit comparable tensile ductility and higher peak stress than the AR800 sample. Tables 4 and 6 also indicate that the proportion between the recrystallized region and the non-recrystallized region is very critical for obtaining good comprehensive mechanical properties. A detailed study on back stress strengthening via partial recrystallization will be investigated in further work.

**Table 6.** Static toughness ($U_T$) of various samples for tension.

| Sample | AH | AR | AR590 | AR660 | AR700 | AR800 |
|---|---|---|---|---|---|---|
| $U_T$ (MJ/m$^3$) | 223 | 27 | 5 | 238 | 280 | 281 |

The above results show that the combined cold-rolling and annealing can not only effectively refine the grains, but also tailor the characteristics and distribution of the precipitated phases. In addition, the annealing temperature also has an important influence on the distribution of microstructures with different characteristics. Therefore, in actual production, a reasonable combination of plastic processing and heat treatment can effectively improve the performance of high-entropy alloys.

## 5. Conclusions

Some conclusions are as follows:

(1) Cold-rolling generates a high density of dislocations and some nano-twins. Such crystal defects promote the precipitation of the B2 phase and induce static recovery and static recrystallization during subsequent annealing. With increasing annealing temperature from 590 °C to 800 °C, the area fraction of the recrystallized region gradually increases from 26.9% to 93.9%.

(2) The (Al, Ni)-rich B2 phase is the dominant precipitate in the AR sample during annealing at various temperatures. The recrystallized region and the non-recrystallized region exhibit distinct precipitation features. The recrystallized region usually contains coarse and equiaxed B2 phases. The fine and elongated B2 phases occupy the non-recrystallized region.

(3) Cold-rolling remarkably enhances the yield strength from 170 MPa to 1060 MPa. Subsequent annealing can largely reduce the work hardening effect via cold-rolling. However, annealed AR samples exhibit far higher yield strength than AH samples. This can be attributed to precipitation hardening via B2 phases, refinement hardening via fine recrystallization, twin-boundary hardening via $\Sigma 3$ twins and residual work hardening in annealed AR samples.

(4) When the temperature is lower than 800 °C, a heterogeneous structure containing recrystallized regions and non-recrystallized regions can be obtained. The proportion between them is dependent on temperature annealing and greatly affects the strength and tensile ductility. It was found that a partially recrystallized microstructure in the AR700 sample exhibits a better combination of strength and toughness than fully recrystallized microstructure in the AR800 sample.

**Author Contributions:** Conceptualization, R.Q. and Q.L.; methodology, R.Q.; formal analysis, R.Q. and B.S.; investigation, Z.Z., T.Y., J.Z. and X.Q.; writing—original draft preparation, Z.Z., R.S. and B.S.; writing—review and editing, R.Q. and B.S.; supervision, R.Q.; project administration, R.Q.; funding acquisition, R.Q. All authors have read and agreed to the published version of the manuscript.

**Funding:** The work was supported by the National Natural Science Foundation of China (Grant No. 51421001), the "111" Project (B16007) by the Ministry of Education and the State Administration of Foreign Experts Affairs of China, project S202010611282 supported by Chongqing Municipal Training Program of Innovation and Entrepreneurship for Undergraduates.

**Data Availability Statement:** The data presented in this study are available on request from the corresponding author.

**Conflicts of Interest:** The authors declare no conflict of interest.

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
