# Peer review of "The Effects of Annealing at Different Temperatures on Microstructure and Mechanical Properties of Cold-Rolled Al0.3CoCrFeNi High-Entropy Alloy"

_metals, doi:10.3390/met11060940_

Round 1
Reviewer 1 Report
I have reviewed the submission "The effects of annealing at different temperatures on microstructure and mechanical properties of cold-rolled Al0.3CoCrFeNi high entropy alloy" by z. Zhu et al. and found some issues that prevent the manuscript to be considered for publication in its original form. The authors may want to follow the given comment in their revision and in that case I may reconsider my decision. My review comments are as follows:
(1) The English writing of the text is not acceptable and seeks careful polishing. There are some sentences that make it difficult to understand the concept.
(2) The sentence "It is found that the amount of recrystallization is largely dependent on the annealing temperature." is trivial to all scholars in the field. These kinds of discussions must be removed as the authors in the abstract and conclusion must present their own findings.
(3) The literature review is weak and needs improvement. The topic of the current research is of interest and there are many recent publications on the HEAs. The following papers may help to improve the introduction part:
[a] Journal of Alloys and Compounds 860, 158412
[b] Metals 10 (12), 1646
[c] Materials Science and Engineering: A 781, 139241
(4) According to tho the design of the experiment in this research, annealing of those samples at <700C was performed for 24h, and for those ones at 700C<T was performed for 10h. It is obvious that time and temperature are both affect recrystallization and in this case, two variables are controlling the ratio of recrystallized material; however, the major part of the given discussion was given on the effect of temperature rather than time. It is expected that the authors give an explanation of their design of the experiment.
(5) The given microscopy studies must be consistent throughout the context. For instance, in Fig. 1, the authors provided the TEM images of only AH and Ar samples and there is no evidence for other annealed specimens using TEM. This is also applicable for EBSD results in Fig. 2.
(6) According to the sentence "Cold-rolling deformation largely promotes the precipitation of B2 phases during annealing. And the characteristics of the precipitates are linked to recrystallization level." the involved mechanisms are not discussed in detail. Why B2 phase precipitation is manipulated by rolling in this HEA?
(7) According to the sentence "Cold rolling generates a high density of dislocations and some nano-twins", the evidence for nano-twins' formation (TEM DP) is not provided.
(8) Does the chemical composition of the B2 phase alter by annealing (the ratio of main elements)?
Reviewer 2 Report
The paper “The Effects of Annealing at Different Temperatures on Microstructure and Mechanical Properties of Cold-Rolled Al0.3CoCrFeNi High Entropy Alloy” addresses an important scientific issue, which is the behaviour of the material after exposure to annealing.
Although the research design is appropriate and the results are clearly presented, there is lack of novelty in this paper in my opinion. Undoubtedly, the authors have access to advanced equipment but repetitively they are stating the obvious – the recrystallized region increases with increasing annealing temperature including all the properties of bigger crystals. Apart from this remark, the Discussion should consider own results and comparison with similar research, not present what other research in this field discovered (e.g. last sentence in the second part of the Discussion). I think that the Discussion needs rewriting so that referenced research if really not necessary for the comparison reasons is moved to the Introduction.
Reviewer 3 Report
The manuscript is devoted to the study of the effect of annealing temperature on the microstructure and mechanical properties of a high-entropy alloy of the AlCoCrFeNi system. In both new and already known high-entropy alloys, a significant improvement in mechanical properties can be realized through microstructural design, achieved not only by searching for optimal combinations of alloying elements but also by selecting rational parameters for deformation processing and heat treatment. Despite numerous studies in the field of synthesis and processing of high-entropy alloys, the issues of the influence of recrystallization during annealing on precipitation have been insufficiently studied, which makes the manuscript undoubtedly relevant. Paper gives a sufficient contribution to the relevant field in accordance with the scope of the journal and the reader’s interest. In general, I believe this manuscript merits publication, and have only a few comments and suggestions to the authors before publication:
1. In the introduction, the authors briefly described the background of the question of the effect of deformation and heat treatment on the precipitation behavior of high-entropy alloys. At the same time, the introduction does not reflect yet many recent relevant fundamental research in the subject area, including those directly related to the subject of the paper (for example, 10.1016/j.matlet.2021.129717, 10.3390/nano11030721, etc.). In particular, it is recommended to supplement the introduction with some results of studies of the strengthening mechanisms of high-entropy alloys.
2. Methods do not provide enough details for the general reader to repeat the experiments. In particular, the technological parameters of arc melting are not given. The purity of the raw materials is not specified (specific percentage of the main element). The brand of the diffractometer is not given and it is not indicated which database was used to decode the diffractograms. It is not specified on what equipment and under what parameters the tensile tests were carried out, etc. It is necessary to describe everything in detail and clearly.
3. It is not clear to what extent statistical sampling mechanical tests were performed (Fig. 8, Tab. 5). There are no confidence intervals, so it is not possible yet to judge the representativeness of these data. Have the results been statistically processed?
4. It is not clear why Fig. 2 shows EBSD maps not for all samples indicated in Table 1.
5. Before conclusions, it is necessary to indicate what practical application the obtained results have and how they can be used in real industrial production.
The publication of the paper seems appropriate after the elimination of the deficiencies identified by the Reviewer.
Reviewer 4 Report
The manuscript titled " The effects of annealing at different temperatures on microstructure and mechanical properties of cold-rolled Al0.3CoCrFeNi high entropy alloy " is presenting the effect annealing temperature of microstructure and mechanical properties in FCC type high-entropy alloy. This paper is not recommended for publication in the Metals, in the present form. The results are interesting and new, but some parts is unclear. Indicates, as follows, one question that need to be improved this manuscript:
1. Did you detected B2 phase and L10 phase bu XRD measurements.
Round 2
Reviewer 1 Report
The authors addressed all my comments and therefore I can recommend publication of the revised manuscript in Metals.